# Optimizing physical scheme selection in RegCM5 for improved air–sea fluxes over Southeast Asia

Quentin Desmet<sup>1</sup>, Marine Herrmann<sup>1</sup>, Thanh Ngo-Duc<sup>2</sup>

<sup>1</sup>Université de Toulouse, LEGOS (CNES/CNRS/IRD/UT3), Toulouse, France

5 <sup>2</sup>REMOSAT Laboratory, University of Science and Technology of Hanoi (USTH), Vietnam Academy of Science and Technology (VAST), Hanoi, Vietnam

Correspondence to: Marine Herrmann (marine.herrmann@ird.fr)

*ORCID:* Quentin Desmet: 0000-0001-7438-5765

Marine Herrmann: 0000-0001-6125-7238

Thanh Ngo-Duc: 0000-0003-1444-7498

10 **Abstract.** This study evaluates the performance of RegCM5 in simulating air–sea fluxes over Southeast Asia through a set of 36 sensitivity experiments testing different physical scheme combinations. Scheme choices vary across five model aspects: radiative transfer, planetary boundary layer, cumulus convection, parameterized microphysics and cloud fraction. A multi-criteria decision-making framework is applied to rank model configurations based on their ability to reproduce spatiotemporal patterns of sea surface wind, latent and sensible heat fluxes, precipitation, and radiative heat fluxes, using 15 mostly satellite-based reference products. No configuration performs consistently best across all criteria: scores assessing latent and shortwave radiative heat fluxes are generally conflicting, with each other and with the scores for precipitation and sea surface wind which instead tend to agree. The choice of cumulus convection scheme drives the performance in simulating the latter two variables, with Tiedtke outperforming and Kain-Fritsch underperforming. In contrast, the best shortwave radiative heat flux simulations are obtained with MIT cumulus convection, in combination with CCM3 radiative 20 transfer. Overall, RRTM/UW-PBL/Tiedtke/SUBEX/Xu-Randall - using the same order of model aspects as listed in the beginning – stands out by maintaining relatively high scores across all assessed variables. Nonetheless, a stronger dissensus in precipitation outputs suggests that reliable rainfall patterns may be a higher priority for decision makers, highlighting CCM3/UW-PBL/Tiedtke/NoTo/Xu-Randall and RRTM/Holtslag/Tiedtke/NoTo/Xu-Randall as the best configurations for this variable. Beyond statistics, further analysis reveals key monsoon-related biases: Indian Summer Monsoon rainfall is generally underestimated, Western North Pacific Summer Monsoon features are overestimated and shifted northward, nearequatorial regions exhibit excessive boreal summer rainfall in most Tiedtke experiments, and austral summer monsoonal sea surface wind and precipitation only impact areas directly north of Australia without inducing the rainfall annual maximum observed in that season over equatorial seas. These findings provide a basis for selecting optimal physics in RegCM5 over Southeast Asia and offer guidance for future applications, including air—sea coupled regional climate modeling.

#### 30 1 Introduction

Southeast Asia (SEA; Fig. 1) faces significant climate instabilities that threaten both long-term sustainability and immediate safety for local populations (about 8.5 % of the world population as of 2023, United Nations, 2024). The region's heavy reliance on climate-sensitive sectors such as agriculture and fisheries makes it particularly susceptible to the impacts of climate change. In the meanwhile, extreme events like tropical cyclones, floods, droughts and wildfires pose constant (McVicar and Bierwirth, 2001; Nguyen-Thanh et al., 2023; Page et al., 2002; Abram et al., 2003; Ummenhofer et al., 2009; Yumul et al., 2011) and growing (Overland et al., 2017; Dagli and Ferrarini, 2019; Eckstein et al., 2019) socio-economic risks.

Figure 1: Annotated bathymetric map of the Southeast Asian domain in our simulations. The eight oceanic subregions employed in the assessment are highlighted with thick colored borders, labeled with their rounded percentage fraction of the total oceanic area in the domain.

Given these vulnerabilities, developing robust adaptation and risk mitigation plans is crucial. Regional climate modeling plays a pivotal role in this endeavor by enhancing our understanding of present and future climate dynamics (Giorgi, 2019). Compared to global models, regional approaches offer finer spatial resolution and tailored parameterizations, thereby enabling a better representation of local processes (Gao et al., 2006; Octaviani and Manomaiphiboon, 2011; Schiemann et al., 2014; Giorgi, 2019). The Southeast Asian Coordinated Regional Climate Downscaling Experiment (CORDEX–SEA) facilitates these efforts in SEA by organizing community experiments in phases ultimately producing updated regional climate projections through dynamical downscaling – i.e. using Regional Climate Models (RCMs) to downscale Coupled Model Intercomparison Project (CMIP) simulations (e.g. Tangang et al., 2020; Herrmann et al., 2022). RegCM (Giorgi et al., 2023) is a major RCM employed by the community, and it is the model of interest in this study.

An absolute prerequisite for conducting accurate regional climate projections is model tuning. With RegCM – and with other RCMs – this generally materializes by conducting sensitivity experiments to the choice of physical schemes throughout several model aspects such as parameterized convection and resolved-scale microphysics. The model typically runs over a period in the near-past when observation datasets are available, enabling an assessment of its performance against them. Best performing physical configurations according to specific criteria (e.g. the reliability of certain variables throughout specific spatiotemporal divisions) can then be selected for conducting optimal future runs. For instance, based on assessments of SEA precipitation by Chung et al. (2018), the CLM land surface model (Oleson et al., 2013) appeared to be more adapted than BATS (Dickinson et al., 1993) in the region. Besides, the ocean surface scheme was found to have only minimal impacts on precipitation and near-surface temperature over land, in terms of both mean and extremes (Juneng et al.,

2016; Cruz et al., 2017; Ngo-Duc et al., 2017), while the study of Li et al. (2016) over both land and sea seemed to indicate a preference for that of Zeng et al. (1998; Eq. (24)). Subsequently, SEA RegCM studies consistently adopted CLM land surface and Zeng et al.'s ocean surface fluxes in the region (e.g. Chung et al., 2018; Zou et al., 2019; Villafuerte et al., 2021; Chung et al., 2023; Ngo-Duc et al., 2024). As for the cumulus convection scheme, Zou et al. (2019) found a better representation of sea surface temperature (SST), rainfall and circulation over East Asia with the Tiedtke scheme (Tiedtke, 1989), although the MIT-Emanuel scheme (Emanuel and Živković Rothman, 1999) appeared more capable to reproduce small scale events – a conclusion consistent with Villafuerte et al. (2021) who focused on tropical cyclones in the Philippine Sea. Lastly, Ngo-Duc et al. (2024) evaluated land precipitation and near-surface temperature for 48 different physical scheme combinations and concluded on the better performance of the Tiedtke cumulus convection scheme when combined with the UW-PBL (Bretherton et al., 2004) and SUBEX explicit moisture schemes (Pal et al., 2000) – although two alternatives using the cumulus convection of Grell (1993) also stood out.

Nevertheless, a notable gap exists in these studies: the performance of RegCM over the ocean has been largely overlooked. Previous sensitivity experiments have primarily focused on key vulnerability variables like precipitation and near-surface temperature over land (e.g. Juneng et al. 2016; Ngo-Duc et al., 2024) – putting aside the more regional studies of Villafuerte et al. (2021) and Zou et al. (2019) which focused on precipitation over the Philippine Sea and East Asia, 75 respectively – and the ability of RegCM to reproduce SEA sea surface conditions was rarely studied, and mostly with fixed configurations (e.g. Herrmann et al., 2020, 2022, who evaluated the ability of RegCM to reproduce the SEA sea surface winds). This is problematic because more than half of the SEA domain is composed of seas (Fig. 1), and many regional climate phenomena, such as monsoons, land-sea breezes and tropical cyclones, are rooted in air-sea exchanges. By neglecting oceanic areas, we thus miss a critical part of the picture. Furthermore, the authors are currently conducting research and development efforts to set up an atmosphere—ocean regional climate model over the region – which aligns with the latest direction of CORDEX suggesting that next-generation regional climate projections will increasingly involve regional Earth system models, notably including air—sea coupling (Giorgi and Gutowski, 2015). To date, the information gap in RCM performance over SEA seas leaves us – and more generally the regional community – largely unprepared for those upcoming coupled modeling projects. While quality reference data may have been lacking in the past, variables such as 85 wind, precipitation and radiative heat fluxes are now well-monitored over the oceans via satellite-based remote sensing, enabling model assessments against these observations. In this study, we aim to fill this gap by assessing the performance of RegCM over SEA seas in terms of air—sea fluxes of heat, mass, and momentum.

RegCM version 5 (RegCM5; Giorgi et al., 2023) is the latest version available today, yet little is known about its capabilities to simulate SEA's climate, particularly with its many newly implemented options (Giorgi et al., 2023, Table 1).

Previous sensitivity studies mostly employed earlier versions such as RegCM3 (e.g. Gianotti et al., 2012), RegCM4 (e.g. Juneng et al., 2016) and RegCM4-NH (Ngo-Duc et al., 2024). Cumulus convection is the aspect gathering the greatest interest (e.g. Cruz et al., 2017; Villafuerte et al., 2021) followed by air–sea fluxes (e.g. Ngo-Duc et al., 2017) and land surface (e.g. Chung et al., 2018), while resolved-scale microphysics, radiative transfer, cloud fraction and planetary boundary layer (PBL) were often set to the defaults. In their last paper, CORDEX–SEA (Ngo-Duc et al., 2024) conducted a few experiments with RegCM5 but, facing unsatisfactory results, called for "thorough testing" before selecting this version for regional climate projections. In response to this call, this article addresses our goal of assessing RegCM5's ability to reproduce air–sea interaction over SEA by conducting sensitivity experiments with its most updated options across various physical scheme combinations.

In this context, we aim to thoroughly evaluate the performance of RegCM5 and its many new physics options in simulating air—sea exchanges over the SEA region. This assessment seeks to provide guidance for choosing optimal physics configurations in future use of the model, including future regional climate projections and air—sea coupling. The remainder of this article is organized as follows. Section 2 provides a description of RegCM5 and details the methodology. Then, Section 3 presents the results. Lastly, Section 4 provides conclusions.

#### 2 Materials and methods

#### 105 2.1 RegCM5

#### 2.1.1 Baseline configuration

RegCM is a distributed regional climate model with a large community base and range of applications, and this study features its most updated version, i.e. RegCM5 (Giorgi et al., 2023). Fig. 1 exhibits the model domain ranging 89.22° E–146.78° E and 18.21° S–27.14° N in longitude and latitude, respectively, such that all SEA seas can be incorporated in the assessment. The horizontal resolution is 25 km – consistent with past CORDEX–SEA studies (e.g. Ngo-Duc et al., 2024). Lateral boundaries' relaxation applies within a lateral buffer area of 24 grid nodes (i.e. around 600 km) following an exponential weighting function (lateral boundary conditions are fully described in Giorgi et al., 1993). Forty-one vertical levels are employed, with an atmosphere top set at 30 km.

We select the MOLOCH non-hydrostatic core, using the staggered Arakawa C-grid (Arakawa and Lamb, 1977) and a height based *ζ* vertical coordinate (Malguzzi et al., 2006; Davolio et al., 2020; Giorgi et al., 2023). The nominal timestep is set to 3 min (advection and sound waves are computed at higher frequency). Timesteps for the cumulus convection (*dtcum*), surface module (*dtsrf*) and radiation (*dtrad*) are set to 6 min, 12 min and 24 min, respectively.

Based on the introduced bibliography, the land surface model and ocean surface scheme are fixed in this study, set to CLM4.5 (Oleson et al., 2013) and to the scheme of Zeng et al. (1998, Eq. (24), which corresponds to the second choice of surface roughness length of momentum in the used implementation of the model), respectively. We disable the parameterization of marine stratocumulus clouds based on Klein and Hartmann (1993; i.e. icldmstrat = 0), which is irrelevant for the domain (Wood, 2012). Moreover, because we aim to explore the full capabilities of the model, we choose not to constrain the model's degrees of freedom regarding cloud formation: convective and large-scale clouds employ distinct liquid water path algorithms (i.e. iconvlwp = 0) and maximum cloud fractional cover settings are set to 100 % (i.e. iconvlwp = 0).

# 2.1.2 Candidate physics options

In Section 2.2 we present the protocol and methods ruling the sensitivity experiments we then conduct, varying the choices made for five model aspects. These five aspects and the candidate options we select are listed in Table 1. The cumulus convection schemes simplified Kuo (Anthes et al., 1987) and Grell (Grell, 1993) are not considered due to their generally poorer performance in RegCM sensitivity studies over SEA (e.g. Sinha et al., 2013; Zhang et al., 2015; Gao et al., 2016; Juneng et al., 2016; Li et al., 2016; Maity et al., 2017). We also exclude the WRF-Single-moment-Microphysics 5-class (WSM5; Hong et al., 2004) option for the resolved-scale microphysics, because preliminary tests with the setups presented in Giorgi et al. (2023) adapted to our domain and resolution did not yield satisfactory results, based on the criteria described in Section 2.2 (not shown). Besides, the NoTo microphysics will only be associated with the Xu–Randall cloud fraction, as done in Nogherotto et al. (2016; spurious cloud amounts appear when combining NoTo with the Sundqvist cloud fraction). As a result, all considered options reported in Table 1 make 36 possible physical scheme combinations.

Henceforth, a given scheme combination in Table 1 will be named after the five-index series corresponding to its five physics option choices. For instance, 12410 refers to the configuration using RRTM for the radiative transfer, the UW-PBL scheme, the MIT cumulus convection scheme, the SUBEX microphysics and the Sundqvist cloud fraction algorithm.

Additionally, a group of configurations sharing identical choices for one or several aspects will be addressed by using asterisks (\*) where the choices are different. For example, \*\*6\*\* configurations are those using the Kain–Fritsch cumulus convection scheme, and 01\*\*1 corresponds to any scheme combination using CCM3, Holtslag and Xu–Randall.

5

| Model aspect                | Selected options                                                                                              |
|-----------------------------|---------------------------------------------------------------------------------------------------------------|
| Radiative transfer          | 0: Modified <b>CCM3</b> (Kiehl et al., 1996)<br>1: <b>RRTM</b> (Mlawer and Clough, 1996; Mlawer et al., 1997) |
| Planetary boundary layer    | 1: Modified <b>Holtslag</b> (Holtslag et al., 1990)<br>2: <b>UW-PBL</b> (Bretherton et al., 2004)             |
| Cumulus convection          | 4: MIT (Emanuel and Živković Rothman, 1999) 5: Tiedtke (Tiedtke, 1989) 6: Kain–Fritsch (Kain, 2004)           |
| Resolved-scale microphysics | <ol> <li>SUBEX (Pal et al., 2000)</li> <li>Nogherotto-Tompkins (NoTo; Nogherotto et al., 2016)</li> </ol>     |
| Cloud fraction              | 0: <b>Sundqvist</b> (Sundqvist, 1988)<br>1: <b>Xu–Randall</b> (Xu and Randall, 1996)                          |

Table 1: Selected physics options for five model aspects. Option indices refer to the corresponding parameter choices in the *physicsparam* namelist of RegCM's input file. Appellations used in the text are highlighted in **bold**.

#### 145 2.2 Protocol and data

# 2.2.1 Simulated period and forcing data

This study focuses on the first mode of variability in SEA, i.e. the seasonal cycle, allowing us to select one single year for assessment (as in, e.g. Ratnam et al., 2009; Gao et al., 2016; Zou et al., 2019, 2020). After assuring that such a period does enable clear discrimination between the simulations' outputs (with appropriate spin up periods), we select the year 2018, which offers rather neutral conditions with respect to large-scale oscillations. Notably, the Indian Ocean Dipole remained neutral throughout the year, except for two slightly positive phases in September and December, and the Southern Oscillations transitioned from a weak La Niña to a moderate El Niño (based on monitoring graphs of the Australian Bureau of Meteorology last accessed 2025-03-26 at <a href="http://www.bom.gov.au/climate/enso/">http://www.bom.gov.au/climate/enso/</a>). The global European Centre for Medium-Range Weather Forecasts (ECMWF) Reanalysis v5 (ERA5, Hersbach et al., 2020), with a horizontal resolution of 1/4° (i.e. around 27 km in the region), 37 vertical levels and a 6 h period, provides initial and boundary conditions for temperature, specific humidity, geopotential and winds.

Then, because our assessment focuses on air—sea fluxes, it is critical to choose appropriate SST forcing data. In particular, submesoscale and mesoscale oceanic processes such as eddies, upwellings and meanders, all considerably shaping the sea surface conditions in SEA (e.g. Da et al., 2019; To Duy et al., 2022; Herrmann et al., 2023), can modulate latent heat fluxes by up to 20 %, thereby affecting cloud formation and precipitation (Frenger et al., 2013; Villas Bôas et al., 2015). The chosen SST forcing data should therefore convey this intense regional sub- and mesoscale activity, at least at and above the scale of 25 km (the horizontal resolution of our model). Optimal-interpolation-based SST datasets (e.g. the Operational Sea Surface Temperature and Ice Analysis, OSTIA, Good et al., 2020, or the Reynolds SST assimilated in the GLORYS12V1 global reanalysis, E. U. Copernicus Marine Service Information [CMEMS], 2020) generally capture the correct spatiotemporal variability on average, but they have been shown to overlook key mesoscale features in the region (To Duy et al., 2022). To address this limitation, our strategy is to force RegCM5 at the sea surface using output from SYMPHONIE (Marsaleix et al., 2008), a regional ocean model that we plan to couple with RegCM in future work as part of an air—sea coupled regional climate system — which underscores the particular relevance of choosing SYMPHONIE for the local long-term strategy. SYMPHONIE benefits from several years of expertise over the region, with a high resolution SEA configuration (Garinet et al., 2024) as well as different configurations throughout the northern SEA (Piton et al., 2021; Nguyen-Duy et al., 2021; To Duy et al., 2022; Herrmann et al., 2023, 2024; Trinh et al., 2024) showing good performances,

in particular with regards to mesoscale processes and sea surface characteristics.. The SEA configuration covers the same domain as in Fig. 1, with a 5 km horizontal resolution and 60 vertical levels, resulting in an effective horizontal resolution of about 20 km, which ensures to resolve oceanic processes in the scale of RegCM's grid. SYMPHONIE runs from January 2017 to December 2018, forced at the lateral boundaries by the GLORYS12V1 global reanalysis, with a horizontal resolution of 1/12° (i.e., around 9 km in the region) and 50 vertical levels (CMEMS, 2020). At the air–sea interface, we employ the COARE3.0 bulk algorithm (Fairall et al., 2003), using the near-surface atmospheric variables of a global analysis from the European Centre for Medium-Range Weather Forecasts (ECMWF; with a 3 h period and at a horizontal resolution of 1/8°, i.e. around 12 km in the region). The resulting SST forcing field shows a good average performance relative to OSTIA (Fig. A1 provides monthly biases to OSTIA through 2018; see also Garinet et al., 2024), and, as expected, exhibits a number of additional mesoscale patterns characteristic of the region. It is employed at the daily scale to force RegCM at the sea surface boundary.

#### 2.2.2 Assessment criteria

We then conduct 36 atmosphere-only RegCM experiments implementing all considered physical scheme combinations (Section 2.1). After a two-month spin up period (the model runs from November 2017), we evaluate these experiments' sea surface fluxes over 2018. Net surface radiative fluxes (longwave and shortwave, hereafter LW and SW) and precipitation (hereafter PR) are compared with satellite data, namely Energy Balanced and Filled (EBAF) Ed4.1 (Kato et al., 2018) and Integrated Multi-satellite Retrievals for Global Precipitation Measurement (IMERG) V07 (Huffman et al., 2023). Latent and sensible heat fluxes (hereafter LH and SH) are compared with the ECMWF analysis that has forced SYMPHONIE. We evaluate sea surface wind (SSW) in comparison with WindSat medium-frequency product v07.0.1 (Wentz et al., 2013; after the comparative study of Hihara et al., 2015). Wind direction is ignored, for great uncertainty/confusion is generally associated with it when conducting spatio-temporal averages. The sign convention of heat fluxes is the following: only SW is positive downward; LW, LH and SH are all positive upward.

The assessment distinguishes between temporal and ocean-only spatial series. Temporal series are monthly annual cycles computed separately over the eight subregions featured in Fig. 1: AND covering the Andaman Sea, SUN for the Sunda continental shelf south of Vietnam, SNS for deep basin of the SEA Northern Sea, JAV for the Java Sea, SCB for the Sulu, Celebes and Banda Seas, SAH for the Sahul continental shelf, and IND and PAC for the Indian and Pacific Oceans, respectively. Designing adequate subregions requires making a compromise between getting reasonably visible cycles (i.e. underlying the fewest seasonality offsets) and not over-dividing the domain as we would then more likely face "overfitting" situations. The subregion partition of Fig. 1 is thus decided according to two criteria: firstly, the atmospheric seasonality of assessed variables, with e.g. regions AND, SUN and SNS being primarily affected by the southwest boreal summer monsoon; and secondly, the coherence of oceanic water masses, notably leading us to match several subregions' boundaries with shelf-break areas (see e.g. SAH which closely follows the Sahul Shelf). Ocean-only spatial series correspond to two mean seasonal patterns, namely December–February and June–August (i.e. the two main seasonal patterns of the year based on Chang et al., 2005; hereafter DJF and JJA). Each spatial and temporal series is assessed against the reference datasets introduced above according to the mean bias (MB), normalized standard deviation (i.e. the standard deviation of the modeled pattern over that of the reference; NSD) and Pearson correlation coefficient (CC), these metrics addressing three distinct aspects of performance, namely, bias, precision and association (based on the nomenclature of Liemohn et al., 2021).

In summary, for each of the six chosen variables, three metrics are employed to evaluate eight subregional annual cycles and two seasonal spatial patterns, which gives a total of 180 criteria. To avoid assessing each scheme combination individually for each criteria, we first rely on a relative performance analysis based on a multi-criteria decision making method (Section 2.3). This method renders a ranking of our experiments, enabling us to select a limited subset of experiments for conducting an absolute performance analysis.

#### 2.3 Multi-criteria decision making

#### 215 **2.3.1 Overview**

We aim to rank our 36 experiments based on their performance regarding 180 criteria. With such a large amount of criteria, it is likely – and it actually is in our case – that no experiment performs strictly better or worse than the others across all objectives. In that respect, multi-criteria decision making serves to seek for "compromise solutions", rather than one absolute best configuration.

The method we present here is derived from Desmet and Ngo-Duc (2021) and was then further utilized (Nguyen-Duy et al., 2023b; Ngo-Duc et al., 2024). Mainly inspired by the Multi-Attribute Utility Theory (MAUT; Keeney and Raiffa, 1976) and the Analytic Hierarchy Process (AHP; Saaty, 1980), it consists of (1) organizing criteria in a weighted hierarchy; (2) applying an exponential utility function to each alternative's rating across each criterion; and (3) computing the experiments' aggregate utility scores using a weighted average of their utilities along the hierarchy.

#### 225 2.3.2 Weighted hierarchy

Figure 2(a) illustrates the weighted hierarchy organizing the criteria assessing one variable in our problem. Variable scores result from three equally weighted subscores featuring the experiments' performance in representing the mean value and the temporal and spatial variabilities. The mean value branch simply aggregates MB results from all patterns, giving at last an equal weight to those calculated from annual cycles and those from seasonal averages. The temporal (spatial) variability branch exclusively considers CC and NSD stemming from annual cycle (seasonal pattern) comparisons. Subregional scores are weighted by the area they represent according to Fig. 1, and subregional/seasonal branches stemming from variability metrics are weighted by the relevant reference pattern's standard deviation (SD).

Figure 2: Schematics of the aggregation process. (a), Illustration of one variable scoring tree. One node (•) represents the average of its entering paths (from above). One tilde (~) stands for the parallel reproduction of a process. (b), Pie chart illustrating the variable weights used for computing the aggregate scores, grouped in three equal slices for the three flux types. The mass flux component of LH's weight is estimated by deriving evaporation from ECMWF's LH then comparing it to PR.

Figure 2(b) then illustrates the weights used for the variable scores at the highest level of the hierarchy. These weights are computed by flux type (heat, mass and momentum are weighted equally) using a representation of the variables' contribution

to these fluxes. Referring to the ocean-only spatio-temporal series from the comparison datasets, we consider the sum of their mean and SD to account for each variable's relative contribution. For example, calculation for the weight  $W_{SW}$  given to SW yields:

$$w_{SW} = \frac{1}{3} \times \frac{\langle s_{SW} \rangle + \sigma_{SW}}{\sum_{v}^{SW,LW,LH,SH}} \langle s_{v} \rangle + \sigma_{v}$$
(1)

where  $S_{v}$  refers to the spatio-temporal series from the comparison dataset used for variable v,  $\langle \cdot \rangle$  denotes an average and  $\sigma$  refers to an SD.

# 2.3.3 Utility functions

Utility functions are computed separately for each criterion. This way, let us focus now on one criterion, i.e fixing j in the equations below. With  $X_{ij}$  the rating of experiment i according to the jth criterion, we first formulate the objective function 250  $f_{ij}$  depending on the employed metric  $m_j$  as follows:

$$f_{ii} = -|x_{ii}| \text{ if } m_i = MB \tag{2}$$

$$f_{ij} = -|x_{ij} - 1| \text{ if } m_j = NSD \tag{3}$$

$$f_{ij} = x_{ij} - 1 \text{ if } m_j = CC \tag{4}$$

such that  $f_{ii}$  is to maximize, with an ideal value of 0. The utility function  $u_j$  can then be applied:

$$u_i(f_{ii}) = \beta_i^{f_{ij}/f_j} \tag{5}$$

where  $f_j$  is the average objective rating among experiments, and  $\beta_j$  is a parameter to calibrate based on the ensemble of objective ratings  $f_{ij}$  along i.

 $eta_j$  drives the exponential slope of the utility function  $u_j$ , thus influencing its discriminating power, which we would like to maximize. To quantify  $u_j$ 's discriminating power, we use the SD along i of  $u_j(f_{ij})$  (i.e the SD of the scores across experiments). With  $eta_j \sim 1$ , the exponential slope is extremely gentle and  $u_j(f_{ij}) \sim 1$  for all i, leading to a low SD of the scores. Conversely, with  $eta_j \gg 1$ , the slope is too steep and  $u_j(f_{ij}) \sim 0$  for all i, again leading to a low SD of the scores. Somewhere between these extremes, there exists an optimal value of  $eta_j$  for which the SD of the scores is maximum, such that on the one hand, some  $f_{ij}$  ratings give near-zero  $u_j(f_{ij})$ , and on the other hand, some others score above 0.5 or near 1. Considering our specific ensemble and criteria, we iterate within (1, 50] to determine the optimal  $eta_j$  for each j. We achieve a precision of this parameter at the second decimal.

# 2.3.4 Aggregation

Lastly, if  $W_j$  refers to the weight of the entire jth branch of the hierarchy, i.e. the vertical product along a branch of all local weights in Fig. 2(a) (provided that the sum of local weights across siblings is always unity), then the aggregate utility score  $U_i$  assigned to the ith experiment is:

$$U_i = \sum_j w_j u_j(f_{ij}) \tag{6}$$

This process can be adapted to each level of the hierarchy, such that each node in Fig. 3(a) can yield a ranking based on the subcriteria it covers. This notably allows to break down aggregate utilities into relevant assessment categories.

#### 3 Results

Based on the protocol described in Section 2.2, we implement 36 RegCM5 physical scheme combinations into experiments, compute statistics assessing their performance at the air—sea interface (all consultable in Appendix B), and conduct a ranking of our ensemble using the multi-criteria decision making technique described in Section 2.3. This ensemble ranking is presented and broken down in Section 3.1. Section 3.2 provides a discussion helping to further interpret this ranking. Section 3.3 scales on these first parts for selecting a subset of experiments, therefore analyzing their outputs beyond statistics.

Figure 3: Scores and associate rankings of the 36 experiments for several stages of the evaluation (all scores are multiplied by 10 for convenience). (a)-(d), Scores by variable (SH and LW scores are not shown) with score contributions from mean value, temporal variability and spatial variability. (e), Scores obtained when aggregating the six weighted variable components: they are grouped by flux type so we can visualize equally weighted contributions from heat, momentum and mass fluxes (LH contributes to both heat and mass flux bars). Each panel in (a)–(e) is topped by a four-rectangle row showing the SD of the panel's scores and each of the three contributing subscores in the same order and color (while the contribution bars correspond to one-third of the subscores such that stacking yields aggregate scores in a 0-10 range, the SD shown on top are all associated with 0-10 scores to enable comparison). (f), Heatmap of the CCs between the score series of each panel - "Ag." standing for "Aggregate scores". Grayscale markers in (a)-(e) refer to the aggregate scores (darker for higher scores in (e)). Simulations are named after the configuration they use, as explained in Section 2.1.

#### 285 3.1 Ensemble ranking

Ensemble ranking is the first step of our protocol, enabling us to statistically analyze relative performance among experiments. Bar charts of the scores are shown in Fig. 3 (we do not show nor discuss the experiments' ability to simulate SH and LW because of their little contribution to the aggregate scores, see Fig. 3(b)). All scores are multiplied by 10 for convenience.

Before pointing out any experiment, let us observe the contribution of the three fluxes to the aggregate scores in Fig. 3(e). These subscores show very different standard deviations (SDs) across the 36 experiments. Quantitatively, SDs of 0.61, 0.89 and 1.21 are obtained for the heat, mass and momentum fluxes, respectively. Great SD indicates that similar rankings have been found throughout the majority of underlying criteria, causing the better members to accumulate good scores while the worse ones cannot stand out, thus increasing the relative performance gap. On the contrary, lower SD 295 indicates a great diversity of the rankings computed throughout the underlying criteria, such that many members see their performance under one criterion offset that under another, thus limiting the relative performance gap observed in the aggregate scores. These two cases are thereafter referred to as "easy-to-rank" and "hard-to-rank" situations, respectively. Following this nomenclature, the heat flux scores' lower SD conveys that the ensemble is harder-to-rank under the heat flux criteria, whereas it is easier-to-rank when dealing with mass and momentum flux statistics. Consequently, mass and 300 momentum flux scores seem to drive the aggregate results, while better scores for the heat flux assessment are seen at all ranks.

While it is tempting to attribute this flux contribution difference to the number of variables they each cover (one for momentum, two for mass, four for heat), it is striking how easy-to-rank the ensemble is when assessing SSW (Fig. 3(d)). The worst and best variable scores as well as the highest SD are indeed obtained for SSW, while all variable scores have been 305 computed from the same number of criteria hence being perfectly comparable in this regard. In particular, Tiedtke configurations (\*\*5\*\*) clearly outperform the others (Fig. 3(d)). The top 11 configurations for SSW indeed exclusively feature Tiedtke experiments, with at least a one score point difference relative to the first non-Tietdke experiment for this variable, namely 11410, ranking 12th (01511 is the only Tiedtke experiment which underperforms, standing at the secondto-last place). As in the aggregate ranking, SSW subscores' contributions from the three assessment aspects, namely mean 310 value and temporal and spatial variabilities, show different spreads. Capacity in reproducing wind temporal variability does not seem to be a discriminating factor (this aspects yields the lowest subscore SD, as shown on top of Fig. 3(d)), whereas reliability of the mean value appears more efficient in that sense, notably being generally responsible for the lower performance of configurations featuring Kain–Fritsch (\*\*6\*\*), behind those using MIT (\*\*4\*\*). Outperformance of Tiedtke experiments however lies in the better quality of their seasonal spatial patterns whose scores simply double between the 12th and 11th members. Coming back to the big picture and temporarily ignoring the unequal variable weights which necessarily modulate the upcoming statements, the higher discriminating power of SSW criteria participates in making SSW scores particularly influential in the aggregate results (Fig. 3(e)). Overall, ranks resulting from the aggregation of two sets of subscores with different SD tend to match those associated with the highest-SD subscores. This is a desirable property, since this procedure dynamically considers the discriminating powers of the input criteria without needing users to make decisions 320 (reducing the amount of subjective choices one needs to make is indeed among the objectives of the ensemble ranking method).

For other variables' subscores (Fig. 3(a)–(c)), mean value is the aspect separating the ten-to-fifteen lower scores from the others. In the upper half of the rankings, assessment of the mean value yields similar results, and score differences stem elsewhere: in the spatial variability branch for PR (though much lighter than for SSW); in the temporal variability branch for SW; and in a combination of both for LH.

LH and SW are the two biggest contributors to the heat flux scores (Fig. 3(a),(b)). Grayscale markers emphasize the low correlations of these variable scores with the aggregate results, equal to 0.63 and 0.37, respectively, relative to about 0.88 for both PR and SSW (Fig. 3(f)). Furthermore, the correlation between LH and SW scores is 0.32: in addition to the fact that the ensemble is harder-to-rank for these variables individually, with score SDs of 0.85 and 0.82, respectively, the two scores tend to offset each other, explaining our first observation on the low influence of the heat flux scores on the final results. Only five configurations rank among the top 10 in both rankings, including notably 02421 and 02410 which both use CCM3, UW-PBL and MIT (024\*\*) and get the highest heat flux contributions to their aggregate scores. Besides, many configurations show very different relative performance at simulating both variables, such as 02610 and several other UW-PBL/Kain–Fritsch experiments (\*26\*\*) which perform much better with SW than with LH.

On the other hand, correlation between PR and SSW scores is 0.78, and many similarities can be observed between both rankings (Fig. 3(c),(d)). We already mentioned the high discriminating power of spatial variability statistics for both variables, but the cumulus convection formulation is also a clear driver of the experiments' performance for PR, and in the same order (i.e. with Tiedtke and Kain–Fritsch outperforming and underperforming, respectively). All configurations of the top seven for PR use Tiedtke, and they are all featured in the SSW top seven as well, although in different order. This way, contrary to the situation with LH and SW, PR and SSW scores rather seem to accumulate, adding to the asymmetry between heat flux contributions and those of momentum and mass fluxes in Fig. 3(e).

Finally, 12511 is the only configuration ranking relatively high for all assessed variables (including when considering SH and LW), its worst rank being fourth for PR, LH and SW. This allows it to rank first overall, even though it never clearly stands out on a variable-wise basis. Changing the Xu–Randall cloud fraction algorithm for Sundqvist (giving 12510, i.e. second aggregate rank) mostly impacts the mean value assessment, yielding lower performance in SW but higher in PR. Besides, 02510 provides the best SSW outputs but shows below-average LH accuracy, 12521 exhibits the best LH performance but is in the second half of the ensemble when assessing SW: each configuration exhibits specific strengths and weaknesses.

# 3.2 Discussion

So far, the applied multi-criteria decision making method allowed us to efficiently characterize our ensemble in terms of relative statistical performance among its members. Yet, this ranking does not hold every information and several points should be discussed.

#### 3.2.1 Impact of the choice of subregions

Firstly, the dependence of the presented ranking to the subregion partition shown in Fig. 1 may be pointed out. To address this point, we experimented alternative partitions with four and six subregions, and the results did not vary significantly (not shown). We are confident in the reasons behind the choice of our final subregion partition as justified in Section 2.2, and thus assure the ranking's robustness in this regard.

# 3.2.2 Pairwise similarity among experiments

Secondly, not all experiment outputs are equally distinct, and equal scores may represent either highly similar or vastly different patterns. This way, since we aim to provide guidance for choosing optimal configurations in future use of the model, one may care about avoiding redundancy or duplication in the case of selecting a subset rather than a single optimal

option. Additional information is thus required for informed decision making, and we employ for this purpose the similarity index of Yamada et al. (2007). This index ranging between approximately zero and one qualifies the consistency in phase, amplitude and mean value within an ensemble of patterns (it can mathematically reach -1 in case of extreme dissensus in the ensemble). As in Desmet and Ngo-Duc (2021), similarity between annual cycles (spatial distributions) is coined "temporal similarity" ("spatial similarity"). In our case, we compute the similarity between each pair of the 36 experiment ensemble for each assessed pattern. While decision makers may consult all the results in Appendix C, we prefer to remain concise here, looking for high similarity pairs among the highest ranking experiments. This way, we first highlight from Fig. C1 and C2 the high resemblance between 12511, 12510 and 02510 (i.e. the overall top three, in order), all sharing the 370 UW-PBL/Tiedtke/SUBEX (PBL/cumulus convection/resolved-scale precipitation) scheme combination (\*251\*). To a lesser extent, high similarity also arises between 01521 and 02521 (ranking fifth and seventh, respectively) which only differ by the choice of PBL scheme. In this context, decision makers could only pick one configuration per similar cluster for constructing a subset of configurations performing well but in a different way, hence reducing the number of considered alternatives. The subset of configurations we further examine in Section 3.3 is constituted using this approach.

#### 3.2.3 Uncertainty on the variable weights

Thirdly, one could fairly question the variable weights employed for the last aggregation. Indeed, even though we have based our variable weights on comparisons of orders of magnitude between the variables of the same category, the one-third factor in Eq. (1) highlights the use of arbitrary equal weights when aggregating results from the three flux components. In fact, how to fairly assess the relative importance of variables? We pointed out that certain criteria had more discriminating power than others, but how does this translate into actual performance differences? The similarity index can be used once again to address this problem, but this time applying it to the entire 36 member ensemble.

Figure 4: (a)-(d), Spatial distribution of the ensemble's temporal similarity index for LH, SW, PR and surface wind. (e), Monthly 385 annual cycles of the ensemble's ocean-only spatial similarity index for LH, SW, PR and SSW.

Figure 4(a)–(d) shows the spatial distribution of temporal similarity for the four variables we focus on, i.e. LH, SW, PR and surface wind (the map in Fig. 4(d) including both land surface wind and SSW). We can note the generally lower similarity along the Equator – quite characteristic of SEA (Ngo-Duc et al., 2017; Desmet and Ngo-Duc, 2021) –, several zones of high 390 dissensus over the SEA seas for LH, and a softer similarity minimum in the northern Philippine Sea for surface wind. For PR, disagreement is particularly more widespread.

This last statement is not apparently contradicted by the results obtained for the similarity of ocean-only spatial patterns throughout the year, which we feature in Fig. 4(e). SW spatial similarity exhibits a clear summer low at around 0.45, contrasting with a winter high ranging from 0.7 to 0.8. LH and SSW show almost no seasonality with a relatively high index of 0.7–0.9. Besides, PR seasonal patterns seem to quite disagree all year long, yielding below 0.6 similarities and reaching minima during the transitional seasons, in April–May and October.

In this context, if one now had to choose one single variable among those we selected to determine a subset of bestperforming experiments in the region, one would likely choose PR – hence emphasizing experiment 02521 (Fig. 3(c)) –
since the ensemble produces the widest range of patterns for this aspect of the climate (i.e. low similarity). Reversely,
without comprehensively examining each SSW outputs, the similarity results for this variable indicate that most experiments
"agree" on the related patterns – whether this generally corresponds to "good" performances or not – thus one would likely
not consider better performances in simulating SSW as relevant configuration-picking criteria. Although this similarity
analysis brings valuable insights into our problem, variable weights remain unclear to define. In this respect, we highlight the
importance of considering each variable ranking individually, and not exclusively consider the aggregate scores which partly
rely on arbitrary weights.

# 3.3 Subset analysis

In the continuity of the performance rankings presented in Section 3.1 then discussed in Section 3.2, a subset of six experiments is selected for a more detailed analysis: 12511 (overall best performer; 12510 and 02510 are ignored based on the pairwise similarities discussed in Section 3.2.2), 12521 (LH best performer and ranking fourth overall), 02521 (PR best performer and ranking seventh overall; 01521 is ignored also based on the pairwise similarity analysis, and prioritizing the analysis of 02521 because of its better performance for PR, which is considered to matter more after our discussion in Section 3.2.3), 11521 (second best performer for PR and ranking sixth overall), 01421 (SW best performer and best MIT configuration overall, i.e. at the eighth rank), and 12621 (second best Kain–Fritsch configuration, i.e. at the 21st rank overall, but best for PR; selecting this experiment allows for a representative of each cumulus convection scheme in the subset, which was a key driver of performance in the performance ranking as seen in Section 3.1).

#### 3.3.1 Precipitation and shortwave radiation

Figure 5 illustrates the subset's spatial performances for PR and SW, taking IMERG and EBAF as references, respectively. Observed seasonal patterns (in DJF and JJA) are displayed in the first row, above bias maps of the six selected experiments. PR and SW subregional annual cycles are shown in the first two rows of Fig. 6.

According to IMERG, a boreal summer monsoon rainfall cycle is observed over AND (May–Aug), SNS and PAC (June–September; Fig. 6). This is coherent with the peak periods of the Indian summer monsoon and Western North Pacific summer monsoon (WNPSM; Wang and LinHo, 2002), and well reproduced by the selected experiments (CC above 0.9 over these regions; Fig. B5), although with a great dissensus on the amplitude. In AND and SNS, the selected subset tends to exhibit dry biases – in conjunction with SW overestimation patterns – along the Indochinese western coast (to a lesser extent with 12511 and 02521), over the northern SNS (particularly for 1\*\*\*21 experiments: this could be partly attributed to the joint use of the RRTM radiative transfer model with the NoTo microphysics) and west of the northern Philippines (Fig. 5). In PAC, although the locally observed PR maximum over the Philippine Sea is not reproduced by RegCM – which is expected regarding our horizontal resolution not designed for modeling extreme events – the WNPSM PR pattern is generally enhanced in our simulations (Fig. 5 and 6). This is particularly true for 12621 and 01421, which additionally exhibit a dry bias in the southern Philippine Sea, hence increasing the contrast with heavy precipitation in the north. 12521 is the selected

experiment showing the most reliable PR pattern over that subregion. Both dry (in AND and SNS) and wet (in PAC) biases are generally also conveyed in the corresponding annual cycles of Fig. 6. However, it is noteworthy that intra-subregion bias compensation – coastal vs. offshore over AND and north vs. south over SNS and PAC – tends to minimize the differences to IMERG curves.

Figure 5: PR and SW seasonal average patterns in IMERG and EBAF reference datasets, respectively (first row), and associated biases for the subset of six experiments (second to seventh rows). Within each of the two great columns (PR on the left and SW on the right), two subcolumns separate DJF (left) from JJA (right) patterns.

Figure 6: Monthly annual cycles of PR, SW, LH and SSW (rows) over the eight subregions (columns), according to the subset of six experiments and to four reference datasets, respectively IMERG, EBAF, ECMWF and WindSat. Gray areas show the envelope of the 36 experiment ensemble mean plus or minus one ensemble standard deviation.

In near-equatorial subregions (SUN, JAV and SCB), IMERG data conveys a PR maximum in austral summer associated with the Australian summer monsoon (Fig. 5 and 6). Yet, Fig. 6 shows that a number of Tiedtke experiments (\*\*5\*\*) produce a boreal-summer-monsoon-type maximum in those regions, contrasting with a dry boreal winter bias. This overly extensive boreal summer monsoon peaks in June–September, May–July and May–August over SUN, JAV and SCB, respectively (Fig. 6). Such deficiency is most starkly verified by 12511, but remains clear with 02521. 1\*521 experiments are less affected by such patterns around the equator (JAV and SCB), and 11521 only exhibits a rainfall bias band in the northern hemisphere, from the Gulf of Thailand (in SUN) to the Philippine Sea (in PAC), while slightly affecting the northern Sulu Sea (in SCB; Fig. 5 and 6). 12521 shares the same summer spatial patterns as 11521 (i.e. changing the PBL scheme does not impact the spatial series' CC), except with a dry tendency. In fact, the model produces most of the austral summer monsoon precipitation in the south of our domain, over SAH and southern IND. 12621 is a good example of this feature, although every selected experiment seems to agree on it (Fig. 5 and 6).

Then, in terms of radiative forcing, observations show that the baseline SW seasonal cycle is primarily driven by the incoming solar flux, as can be seen in Fig. 6 during periods of low PR variability. This way, SW over SAH and IND (in the southern hemisphere) gradually increases during the local dry season from June to November, toward the austral summer maxima. Conversely, a January-to-May SW increase is very clear in the northern hemisphere over SNS and PAC, while PR stagnates around its yearly minima. Those transitional season SW trends are overall well reproduced by RegCM, although with a positive (negative) bias over SNS (IND and PAC). Over SAH, the ensemble is unbiased during the dry season, and shows little spread.

Besides, near-equatorial subregions (SUN, JAV and SCB) experience a double peak in their SW annual cycle, with maximum values during transitional seasons and minimum values in winter and summer (Fig. 6). The over-extended boreal summer monsoon signals commented above strongly impact this baseline seasonality. Over SUN, temporal SW NSD for 12511 and 02521 exceed 1.8 (among the highest figures for this statistic and variable among the eight subregions, as can be visualised in Fig. 6; explicit numbers are also shown in Fig. B1), as dry (wet) biased conditions in boreal spring (summer) enhance the observed SW maxima (minima). On the other hand, the lowest temporal CC are obtained over JAV, with 0.25–470 0.6 for the selected experiments apart from 11521 and 12621 which give above 0.7 CC (see complete statistics in Fig. B1). It is worth noting that this disrupted SW and PR seasonality occurs over the shallow seas of the continental shelf and thus might strongly affect the heat and water budgets of the oceanic components in the perspective of air—sea coupling.

Finally, in spite of a generally inverse correlation between PR and SW patterns – which can be explained by the radiative effect of convective clouds – unbiased SW is rarely modeled together with unbiased PR, and two different configurations do not systematically associate similar PR biases with similar SW biases. For instance, even though 01421 shows a drier boreal summer than 12621 around the equator, the former experiment appears particularly reliable in this same region for SW, while the latter exhibits positive biases (Fig. 5). Meanwhile, comparably wet biases during the same season in 02521 and 12511 are associated with a widespread negative SW bias for the former, whereas this level of difference is reached only locally for the latter. Thus, in addition to the high dissensus among the ensemble members for modeling JJA spatial distribution of both variables (Fig. 4(e)), there is great variability in their relationship. This highlights a disagreement in RegCM5 between the radiation and precipitation (whether cumulus convection or resolved-scale microphysics) schemes on the level of cloudiness that must be associated with satisfactory SW inhibition and rainfall.

#### 3.3.2 Sea surface wind speed and latent heat

Figure 7 illustrates the subset's spatial performances for LH and SSW, taking respectively ECMWF and WindSat as references and displaying seasonal observed patterns and bias maps for the six selected experiments. LH and SSW subregional annual cycles are shown in the last two rows of Fig. 6.

Figure 7: As in Fig. 5, but for LH and SSW, taking ECMWF and WindSat as references, respectively.

Similarly to PR and SW, LH and SSW are closely related, although in this case the correlation is positive (Fig. 6 and 7). This is directly explained by thermodynamics principles, evaporation being enhanced with increased SSW by construction, yet this correlation is also modulated by several factors such as near-surface temperature and humidity. That leads RegCM to simulate various LH responses to a given SSW pattern depending on the configuration used. One can for instance take notice of the different ensemble spreads for LH between the annual cycles of JAV and SAH (Fig. 6), whereas there is similar agreement between the simulations when modeling SSW over those subregions. This highlights the relevance of evaluating both LH and SSW. The observed seasonality of both variables is very well reproduced by the selected experiments (Fig. 6). We thus conduct the following analysis by focusing on the spatial performance (Fig. 7), using the seasonal cycles (Fig. 6) as

535

For every experiment, a DJF positive SSW bias is produced in the eastern South Indian Ocean–Timor Sea, ranging from +1 m s<sup>-1</sup> ( $\sim$ 16 %) with 12511 to +3 m s<sup>-1</sup> ( $\sim$ 50 %) with 12621 (Fig. 7). Except for 11521, it is associated with intense LH biases, reaching up to +80 W m<sup>-2</sup> (~50 %) in 12621. SSW seasonal cycles over SAH (Fig. 6) show that the 505 overestimation during December–January is consensual. The bias variability among experiments acknowledged in Fig. 7 for DJF in SAH is thus mainly rooted in February (Fig. 6), as all experiments except 01421 and 12621 (which maintain overestimation in February) are unbiased for this month's subregional mean, hence partly compensating for the previous months' overestimation.

Conversely, in boreal summer, the local LH maximum observed in the Timor Sea is underestimated in many RegCM runs, with a small to intense negative bias extending further to the west in the Indian Ocean (Fig. 7). At most, a -80 W m<sup>-2</sup> (-40 %) bias is obtained in 01421 and 1\*521 experiments. This LH underestimation is generally not associated with SSW underestimation. In the same season, north of that LH underestimation, both LH and SSW are often overestimated around the equator or slightly to its south, affecting IND, JAV and SCB (Fig. 7). In terms of LH, this overestimation 515 represents a relatively narrow band not exceeding +40 W m<sup>-2</sup> with 12521; it is widely spread around the equator and remains under +50 W m<sup>-2</sup> with 12511; and it even reaches +70 W m<sup>-2</sup> with 12621 (from +15 to +70 % depending on the area). Together with the LH underestimation in the south, these patterns suggest a northward shift/extension of the high LH and SSW zone visible in the reference panel along the domain's southern boundary – corresponding to the dry lower branch of the winter hemisphere's Hadley Cell. In terms of SSW, a +1 to +2 m s<sup>-1</sup> overestimation is generally associated with this LH 520 overestimation pattern. However, while similar SSW overestimation is obtained for 12521 and 11521, the latter exhibits no collocated LH peak (Fig. 7). This could be attributed to the PBL scheme, since 11521 uses Holtslag whereas experiments overestimating LH typically use UW-PBL (\*2\*\*\*). With the current parameters, Holtslag could induce less turbulence than UW-PBL near the surface hence affecting \*1\*\*\* simulations' capacity to extract moisture from the ocean. On the LH annual cycles of Fig. 6, the two lowest curves for IND, JAV and SCB are indeed the two \*1\*\*\* simulations of the subset. Over 525 IND, these simulations are lower than ECMWF's cycle, because the southern negative bias is not compensated as with \*2\*\*\*. Over JAV and SCB, Holtslag experiments are closer to the reference.

SSW overestimation during the WNPSM (i.e. during JJA, over SNS, and the northern Philippine Sea in PAC) is also frequent, but only 01421 and 12621 (the two non-Tiedtke experiments) bring it to levels significantly impacting LH 530 (Fig. 7). In particular, 01421 increases SSW by nearly +70 %, from  $\sim$ 6 m s<sup>-1</sup> to  $\sim$ 10 m s<sup>-1</sup>, which is a value not even observed in the region during winter when trade winds reach the SNS. As a result, 01421's curve is a clear outlier over SNS (Fig. 6). Over PAC, one can note in Fig. 7 additional negative biases in the southern Philippine Sea for both experiments (compensating for the northern positive biases when spatially averaging for the annual cycle). This is consistent with the PR bias patterns previously discussed in Section 3.3.1, in association with a generally enhanced WNPSM.

Lastly, Fig. 7 highlights a number of differences between RegCM simulations and ECMWF's LH spatial distribution, consistent throughout all selected experiments: mesoscale spatial oscillations in the northern Philippine Sea in DJF (particularly seeable with 12621), as well as negative bias around Taiwan in DJF and curved negative bias pattern offshore South Vietnam in JJA. These differences coincide with zones of upwelling and/or mesoscale eddies and meanders 540 which result in locally cold SST biases in SYMPHONIE when compared with OSTIA (Fig. A1). This mesoscale activity is

certainly not accounted for in ECMWF, thus those biases cannot be attributed to RegCM. Since these patterns are shared by most experiments, they do not influence the relative performance assessment we conducted in Section 3.1.

# 4 Synthesis and conclusion

This study proposed to optimize the choice of physical schemes in RegCM5 for the best performance of the model at the sea surface, seeking to provide guidance for selecting appropriate physics configurations in future modeling experiments (notably, involving air—sea coupling). Employing among the most recent schemes released in the last version of the model, 36 scheme combinations were implemented varying the choices made for five model aspects: radiative transfer, planetary boundary layer (PBL), cumulus convection, resolved-scale microphysics, and cloud fraction. We then evaluated for the year 2018 those RegCM5 experiments' ability to reproduce regional sea surface fluxes of heat, mass and momentum.

Precipitation and shortwave and longwave radiative heat fluxes as well as sea surface wind speed were compared to satellite data, and latent and sensible turbulent heat fluxes were evaluated against an analysis. Eight oceanic subregions were designed for computing subregional average annual cycles of the studied variables, based on the joint consideration of bathymetric features and known subregional seasonality. Spatial distribution patterns were assessed in terms of seasonal means. The correlation coefficient, normalized standard deviation and mean bias were the metrics used to compare the 555 experiments' temporal and spatial series to reference data, allowing to differentiate errors of phase, amplitude and mean value, respectively. Deriving the method of Desmet and Ngo-Duc (2021), we organized the 180 criteria into a weighted hierarchy, then applied an adaptive utility function hence yielding scores and rankings at each level of the hierarchy. This multi-criteria decision making approach proved powerful in informing about the relative performance across criteria among the 36 experiment ensemble.

The ranking results emphasized several top performing RegCM configurations:

- 12511 (using RRTM/UW-PBL/Tiedtke/SUBEX/Xu-Randall) and 12510 (changing Xu-Randall for Sundqvist) ranked first and second overall respectively, 12511 being the only configuration ranking in the top five for every variable ranking;
- 02510 (using CCM3/UW-PBL/Tiedtke/SUBEX/Sundqvist) ranked third overall and first for SSW;
- 12521 (changing SUBEX for NoTo, in comparison with 12511) ranked fourth overall and first for LH;
- 01521 (changing RRTM for CCM3, in comparison with 12521) ranked fifth overall;
- 11521 (changing UW-PBL for Holtslag, in comparison with 12521) ranked sixth overall and second for PR;
- 02521 (changing Holtslag for UW-PBL, in comparison with 01521) ranked seventh overall and first for PR;
- 01421 (changing Tiedtke for MIT, in comparison with 01521) ranked ninth overall and first for SW.

The majority of the top 10 scheme combinations featured the Tiedtke cumulus convection, the UW-PBL, and the Xu-Randall cloud fraction schemes, while representing each option for radiative transfer and resolved-scale microphysics nearly equally. The Tiedtke cumulus convection scheme particularly outperformed others for modeling precipitation and sea surface wind. Conversely, Kain-Fritsch experiments exhibited the lowest ranks on average, the best simulation standing 20th overall. No configuration was found to perform consistently better across all variables (12511 ranked relatively high in each variable ranking, but only ranked first after aggregating the variable scores). In particular, while many Tiedtke configurations appeared to perform well for both precipitation and sea surface wind, the three first-ranking experiments for shortwave radiative flux used MIT, and their scores for this variable showed a clear performance gap with 12511 ranking fourth. Besides, a similarity analysis (Section 3.2.3) informed about the enhanced variety of patterns among the ensemble for precipitation when compared to other variables yielding less dissensus, which suggests that a better performance at simulating precipitation might matter more to decision makers – although to what extent is uncertain.

Ignoring several well performing experiments which showed a high level of similarity with each other (based on Section 3.2.2), we then constituted a subset of six top-performing configurations for an absolute performance analysis, 585 helping to characterize the experiments' patterns beyond their scores and statistics. The major conclusions of the subset analysis are summarized as follows. RegCM5 captures reasonably well the seasonality of sea surface wind, latent heat flux,

as well as shortwave radiative heat flux under low-rainfall conditions. The timing of summer and winter monsoons is overall well captured, although their intensity and spatial extent are modeled with a great dissensus: boreal summer near-coastal rainfall in the northwest is generally underestimated (especially in RRTM-NoTo experiments); Western North Pacific 590 Summer Monsoon rainfall, sea surface wind, and latent heat flux are overestimated and shifted northward (particularly in non-Tiedtke experiments); in near-equatorial regions, most Tiedtke experiments produce an unrealistic annual rainfall maximum in boreal summer instead of boreal winter; Australian Summer Monsoon precipitation, sea surface wind, and latent heat flux are intensified over the continental shelf north of Australia, which coincides with an austral summer dry bias in near-equatorial seas. UW-PBL experiments tend to simulate higher latent heat flux than Holtslag ones, attributed to 595 enhanced turbulence.

Several limitations are yet to be mentioned. Indeed, our strategy targeted the seasonal cycle, assessed over a single year selected for its neutrality with respect to large-scale oscillations. Consequently, our findings are specific to that context, and no conclusions can be drawn about the performance of this study's top-performing configurations under non-neutral conditions, in terms of intraseasonal/inter-annual variability or more. While this work provides a basis for identifying a subset of promising configurations, additional experiments are needed to further refine the selection, notably involving longer simulations to conduct more comprehensive diagnostics.

Also, the analysis of spatial distributions highlighted bias compensation within subregions, rooted, e.g., in coastal vs. offshore performance contrasts (although the robustness of the ranking to the choice of subregion partition was validated 605 in Section 3.2). Since such bias compensation depends on experiments, it appears difficult to select a perfectly relevant subregion partition, but we could use more systematic protocols such as clustering techniques hence making the selection less arbitrary. Taking a broader view, this also questions the definition of criteria by fixed spatiotemporal subdivision - i.e. by subregion and by season – while the climatic features shaping the studied patterns (e.g. the monsoons) are moving in time and space. In this respect, process-oriented diagnostics could be more appropriate and should be explored in future studies.

Moreover, it is worth reminding that all the physical schemes tested throughout this paper were employed with their default internal parameters. Regarding the improvements obtained in the model of Zou et al. (2014) after fine-tuning their cumulus convection scheme, one could fairly question the relevance of our study. Indeed, although not explicitly mentioned, our work relied on the disputable hypothesis that our schemes' internal parameters are all relatively close to their "tuned" value. Based on the analysis of our top-performing experiments in Section 3.3, where, in particular, unbiased precipitation 615 was never associated with unbiased surface shortwave fluxes, we can confidently state that this hypothesis is not valid. In addition, the "tuned" values of, e.g., the Tiedtke cumulus convection scheme, would certainly differ whether the scheme is employed in combination with, e.g., for the resolved-scale precipitation scheme, SUBEX or NoTo, or when varying any other physical option in the model. So, ideally, a more thorough protocol would have been to fine-tune the internal parameters of the five schemes chosen altogether, and to repeat the process for every different scheme combination, i.e. 36 620 times. Only then could we apply our strategy to assess and rank the 36 fine-tuned configurations based on their performance at the air-sea interface, select a subset for a more thorough analysis, etc. Such work would however have involved considerably more computing resources and time than we could afford. Keeping the default internal parameters was therefore only pragmatic.

Another approach within the same constraints would have been to fine-tune one single configuration we would have 625 chosen based on previous research. However, previous studies featured significantly less physical options than those tested here, and focused almost exclusively on land performance. For example, Ngo-Duc et al. (2024) recently employed 0\*\*\*0 configurations to assess land temperature and precipitation (and to our knowledge, only 01\*10 experiments were tested in earlier works). As a result, our understanding of how RegCM performs across the full SEA domain was incomplete, and some recent options were never assessed despite yielding good results in the present study (e.g. RRTM and Xu-Randall). 630 After the current paper, assessing RegCM's most updated schemes over land would be a valuable follow-up. Nonetheless, in

order to guide modelers seeking homogenous RegCM performance over the region, we can conduct as of now a brief comparison of our ocean-focused results with the land-only ones of Ngo-Duc et al. (2024). They notably identified four configurations with equivalent aggregate scores, including three using Kain–Fritsch and one using Tiedtke. Our results indicate that Kain–Fritsch tends to overestimate oceanic monsoon signals in terms of precipitation, sea surface wind and latent heat flux, such that Kain–Fritsch configurations generally ranked in the bottom third of the ensemble. This supports favoring their top experiment that used Tiedtke instead. The Tiedke configuration highlighted in their study (i.e. 02510 using our notation) ranked third overall in ours while sharing the same PBL (UW-PBL), cumulus convection (Tiedke) and microphysics (SUBEX; i.e. \*251\*) as in the first and second ranks. This suggests that a balanced configuration may lie among these \*251\* combinations. Our work thus serves as a prerequisite before embarking on any fine-tuning efforts from a relevant configuration. According to our findings, future fine-tuning efforts should first target the cumulus convection scheme, which was the primary driver of performance. Radiative transfer, PBL, and microphysics should follow as secondary priorities, while the cloud fraction algorithm warrants lower focus.

The research proposed in this article also invites further exploration. For example, while we chose to force RegCM with a high-resolution SST field from SYMPHONIE in place of traditional, smoother SST datasets, we did not address the impact of this choice on the outputs of the model. How oceanic mesoscale eddies and meander impact the formation of clouds and precipitation in the area? We employed a 25 km horizontal resolution, so this may limit the impact of oceanic mesoscale in the atmosphere, but will this influence of SST become more critical with future resolution improvements? Indeed, with the upcoming seventh phase of the Coupled Model Intercomparison Project (CMIP; Dunne et al., 2024), resolution should increase in both global and regional climate models (including those we employ). More generally, how will our performance ranking evolve with those new resolutions?

Lastly, we want to ensure the relevance of our results in the perspective of using RegCM as the atmospheric component of a coupled system involving air—sea coupling. Indeed, referring to Hourdin et al. (2017), tuning a coupled modeling system usually requires several stages, including "component tuning" with the standalone components, before conducting a "system tuning" with the fully coupled setup. Moreover, in their review on air—sea coupled modeling over the Maritime Continent, Xue et al. (2020) pointed out that uncertainties in modeling convective processes, fractional coverage and autoconversion — all strongly affecting surface fluxes — are primarily driven by their formulation and associated parameters within the atmospheric component, rather than by ocean—atmosphere coupling. In that respect, the insights gained from our atmosphere-only experiments provide valuable guidance for future air—sea coupled simulations. Such modeling perspectives are a key focus of the authors' ongoing research and development efforts.

# Appendix A

Figure A1: 2018 monthly biases between the SST forcing data used in RegCM5 (issued from a high resolution simulation from the regional ocean model SYMPHONIE) and OSTIA's analysed SST. Note that OSTIA's analysis error in SEA is about 0.2 °C far from land, and ranges between 0.5 and 1 °C in near-coastal areas. In each subplot's title, "av" and "sd" show the average and standard deviation of the displayed bias field, respectively.

# SW (17 %)

Figure B1: Raw statistics assessing the 36 experiments' SW fields, serving as a basis for the ranking shown in Fig. 3. The percentage in the title recalls the variable weight of Fig. 2(b). Below, the bar graph indicates the relative weights among the 30 criteria across columns. Then, the heatmap exhibits the raw statistics across experiments (rows) and criteria (columns). Three different colormaps are employed for three different metrics, all indicating brighter shades for better performances: from dark red to white for CC; from dark green to dark purple for NSD (with white assigned to NSD=1); and from dark blue to dark red for MB (with white assigned to MB=0). The colormaps' scaling is computed independently for each column, based on local minimum and maximum values across rows (for NSD and MB, the scaling is symmetric, based on the maximum distance to 1 and 0, respectively). Experiments are displayed in arbitrary order. The six experiments included in the subset of Section 3.3 are emphasized in bold.

Figure B2: As in Fig. B1, but for LW.

Figure B3: As in Fig. B1, but for SH.

Figure B4: As in Fig. B1, but for LH.

Figure B5: As in Fig. B1, but for PR.

Figure B6: As in Fig. B1, but for SSW.

Figure C1: Pairwise spatial similarity indices across the 36 experiments and two seasons. Similarity indices are applied to DJF (JJA) patterns in the upper-left (lower-right) section of each subplot. The cells corresponding to the three pairs one can constitute with 12511, 12510 and 02510 are framed in orange.

Figure C2: As in Fig. C1, except with the temporal similarity indices across the eight oceanic subregions. Similarity indices are applied to SUN, JAV, SAH and AND (SNS, SCB, PAC and IND; displayed clockwise within each cell starting at the top) patterns in the upper-left (lower-right) section of each subplot.

# 705 Code availability

The RegCM5 code version used for this work is produced by the Abdus Salam International Centre for Theoretical Physics (ICTP, 2025) and can be accessed at <a href="https://doi.org/10.5281/zenodo.15125814">https://doi.org/10.5281/zenodo.15125814</a>. A Python package was also coded to implement the ranking logic although it is not ready for distribution. The version used in this paper can be accessed at <a href="https://doi.org/10.5281/zenodo.15356967">https://doi.org/10.5281/zenodo.15356967</a>. Last, miscellaneous scripts to reproduce all the data and visuals presented in this paper can be accessed at <a href="https://doi.org/10.5281/zenodo.15359231">https://doi.org/10.5281/zenodo.15359231</a>.

# **Author contribution**

QD contributed to conceptualization, data curation, formal analysis, investigation, methodology, software, validation, visualization and writing. MH and TND contributed to conceptualization, funding acquisition, investigation, project administration, resources, supervision and writing – review & editing.

# 715 Competing interests

The authors declare that they have no conflict of interest.

#### Acknowledgments

The editor and the authors sincerely thank two anonymous referees for their time and effort in providing peer reviews of this manuscript. This work was granted access to the HPC resources of CALMIP supercomputing center under the allocation 2023-p20055. In the figure-making process, we thankfully employed the Tableau10 color palette as well as the perceptually uniform colormaps from the "cmocean oceanographic colormap package" and the "scientific colour maps" developed by Fabio Crameri.

# Financial support

This study is supported by the LOTUS International Joint Laboratory (<a href="http://lotus.usth.edu.vn">http://lotus.usth.edu.vn</a>) funded by the French
National Research Institute for Sustainable Development (IRD), and is a contribution to the SEASTERS project funded by
CNES. TND is supported by the Vietnam National Foundation for Science and Technology Development (NAFOSTED)
under Grant 105.06-2021.14.

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
