# Peer review of "Optimizing physical scheme selection in RegCM5 for improved air–sea fluxes over Southeast Asia"

_EGUsphere, 2025_

## Referee Comment (RC2)

**Review of the manuscript:**

**"Optimizing physical scheme selection in RegCM5 for improved air–sea fluxes over Southeast Asia"**

by *Quentin Desmet, Marine Herrmann, Thanh Ngo-Duc*

In this manuscript, the authors evaluate the performance of the RegCM5 regional climate model in simulating air–sea fluxes over the Southeast Asian seas. The model was run at a 25 km resolution for the year 2018 using 36 different combinations of physical parameterization schemes, selecting from multiple options for convection, microphysics, planetary boundary layer (PBL), radiation, and cloud fraction. Atmospheric forcing was provided by ERA5 reanalysis at 0.25° resolution, while sea surface temperatures were obtained from the high-resolution SYMPHONIE ocean model running at approximately 0.083° resolution. Model outputs, such as precipitation, surface radiation, latent and sensible heat fluxes, and sea surface wind speed, were evaluated against satellite and reanalysis data. A multi-criteria decision-making framework, incorporating 180 performance metrics across eight oceanic subregions, was used to rank the experiments. The results indicate that the top-performing configuration is a combination of the RRTM radiative transfer scheme, UW-PBL planetary boundary layer, Tiedtke cumulus convection, SUBEX resolved-scale microphysics, and Xu–Randall cloud fraction (identified as 12511, i.e. RRTM/UW-PBL/Tiedtke/SUBEX/Xu–Randall), with the Tiedtke cumulus convection scheme consistently outperforming others, particularly in simulating precipitation and wind. The findings highlight cumulus convection as the primary driver of model performance and suggest that the optimal physical parameterizations may vary depending on the variable of interest (e.g., precipitation vs. shortwave radiation). The manuscript is well written, logically structured, and easy to follow, making it a worthy candidate for publication in Geoscientific Model Development. However, there are some points need to be clarified.

1) First, the authors' use of simulation results from only one neutral year (2018) to evaluate the model's performance is not sufficiently convincing. A single-year simulation provides only one monthly and annual value per grid point for each variable, which introduces substantial uncertainty into the performance assessment due to the lack of statistical robustness. Furthermore, by excluding years influenced by major climate variability phenomena such as ENSO and IOD, the evaluation overlooks the model's capacity to simulate responses under extreme conditions, one of the key strengths of dynamical models. As a result, the findings may be overfitted to neutral conditions and may not adequately reflect the model's robustness or broader applicability across different climate regimes.

2) This study is highly valuable for advancing our understanding of air–sea coupling and for supporting the development of coupled models. However, in many practical applications, the accurate simulation of precipitation and temperature over land is even more critical. In fact, coupled models are still relatively uncommon, and most studies continue to rely on standalone RegCM without ocean coupling. Therefore, I suggest that the authors conduct a parallel analysis using the same model configurations over terrestrial subregions where high-quality observational data are available.

3) Using ERA5 at the same resolution (0.25°) to force RegCM5 is valid and appropriate for a controlled physics sensitivity study, as done by the authors. However, in this setup, the added value of high-resolution spatial detail from the regional model cannot be fully realized./.

---

## Author Comment (AC1)

**General overview**

*The authors evaluate the performance of RegCM5 in simulating air-sea fluxes over the Southeast Asian region by testing different physical parametrizations: radiative transfer, planetary boundary layer, cumulus convection, parameterized microphysics, and cloud fraction. To assess the performance of numerical simulations, the authors use a multicriteria decision-making framework to rank their ability in reproducing sea surface wind, latent and sensible heat fluxes, precipitation, and radiative heat fluxes. The authors have not found a configuration that properly solves all criteria. However, they found that using the Tiedtke cumulus convection scheme showed better results for precipitation and sea surface wind.*

We thank you for your overview, as well as your analyses and comments. We restructured your text by topic to make it easier to address. Accordingly, you can find our answers below (line numbers refer to the updated manuscript).

**Comments**

**1. About the usefulness of resolution:** *The authors mention that the study focuses on the seasonal cycle and that they have chosen the year 2018 to perform the numerical simulations since it offers neutral conditions with respect to large-scale oscillations. However, the authors remark that it is important to properly take into account the influence of upwellings, eddies, and meanders when forcing atmospheric models. This is the reason why they force RegCM5 with a high-resolution regional SYMPHONIE forced numerical simulation (5 km) instead of with an optimal-interpolation-based SST dataset. With a 5 km spatial resolution, the ocean numerical simulation will be able to solve eddies of at least 20 km in diameter (effective spatial resolution). When forcing the atmospheric model, the sea surface data, containing high-spatial resolution features, will be interpolated to the 25 km grid of RegCM5, passing from an effective spatial resolution of 20 km to 100 km.*

We appreciate the attention brought to resolution here. We however only partially agree with the flow you propose. Indeed, it is correct and intended that the effective resolution of our ocean numerical simulation is about 20 km, standing a bit below the target resolution of 25 km for the forcing, but interpolating onto this coarser grid does not reset the effective resolution computation, bringing it to 100 km as you suggest, since the ocean is not modeled again during the atmospheric run. With a 5 km grid, SYMPHONIE resolves 20 km scale processes, these processes' SST signature affects the interpolated values at 25 km for forcing RegCM, and this is all. For the only aspect of forcing data resolution, this is, in our opinion, the best we can do regarding our atmospheric grid.

In any case, the updated manuscript now contains some words about this effective resolution aspect you fairly pointed out:

L162 "The chosen SST forcing data should therefore convey this intense regional sub- and mesoscale activity, at least at and above the scale of 25 km (the horizontal resolution of our model)."

And a little bit further:

L175 "[...] resulting in an effective horizontal resolution of about 20 km, which ensures to resolve oceanic processes in the scale of RegCM's grid."

**2. On the role of mesoscale processes:** *At the end, there is no discussion about the role of mesoscale processes, such as upwelling, eddies, or meanders, in modulating heat fluxes, precipitation, or winds. Following Frenger et al. (2013) and Villas Bôas et al. (2015), cold and warm eddies are important drivers of latent heat fluxes, and because of that, can modulate the cloud cover and precipitation. In fact, Villas Bôas et al. (2015) showed that eddies can partially modulate 20% of latent heat fluxes.*

The role of mesoscale processes in modulating the sea surface fluxes is indeed a valuable topic that connects with our paper. While it is not included in the scope of the current paper – primarily focusing on selecting optimal schemes – developing such questions is somehow planned for our future research, in particular using air–sea coupled regional climate modeling. For the current study, all simulations employ the same SST forcing, such that the specific effects of oceanic mesoscale on the surface fluxes should not affect our conclusions (i.e., how we compare configurations with each other), as partially mentioned when commenting on the latent heat seasonal maps all similarly conveying specific mesoscale patterns rooted in the SST field forcing:

L538 "Lastly, Fig. 7 highlights a number of differences between RegCM simulations and ECMWF's LH spatial distribution, consistent throughout all selected experiments: mesoscale spatial oscillations in the northern Philippine Sea in DJF (particularly seeable with 12621), as well as negative bias around Taiwan in DJF and curved negative bias pattern offshore South Vietnam in JJA. These differences coincide with zones of upwelling and/or mesoscale eddies and meanders which result in locally cold SST biases in SYMPHONIE when compared with GLORYS (Fig. A1). This mesoscale activity is certainly not accounted for in ECMWF, thus those biases cannot be attributed to RegCM. Since these patterns are shared by most experiments, they do not influence the relative performance assessment we conducted in Section 3.1."

Moreover, we thank the referee for the provided references, which we think help to frame how critical it is to choose appropriate SST forcing data. The comment led to a refinement of the following paragraph:

L159 "Then, because our assessment focuses on air–sea fluxes, it is critical to choose appropriate SST forcing data. In particular, submesoscale and mesoscale oceanic processes such as eddies, upwellings and meanders, all considerably shaping the sea surface conditions in SEA (e.g. Da et al., 2019; To Duy et al., 2022; Herrmann et al., 2023), can modulate latent heat fluxes by up to 20 %, thereby affecting cloud formation and precipitation (Frenger et al., 2013; Villas Bôas et al., 2015). The chosen SST forcing data should therefore convey this intense regional sub- and mesoscale activity, at least at and above the scale of 25 km (the horizontal resolution of our model)."

A new paragraph opening up the perspectives of the study now includes this consideration (together with ideas of the other anonymous referee):

L646 "The research proposed in this article also invites further exploration. For example, while we chose to force RegCM with a high-resolution SST field from SYMPHONIE in place of traditional, smoother SST datasets, we did not address the impact of this choice on the outputs of the model. How oceanic mesoscale eddies and meander impact the formation of clouds and precipitation in the area? We employed a 25 km horizontal resolution, so this may limit the impact of oceanic mesoscale in the atmosphere, but will this influence of SST become more critical with future resolution improvements? Indeed, with the upcoming seventh phase of the Coupled Model Intercomparison Project (CMIP), resolution should increase in both global and regional climate models (including those we employ). More generally, how will our performance ranking evolve with those new resolutions?"

**3. Why forcing with SYMPHONIE?** *When the authors compare the monthly SYMPHONIE sea surface temperature results with OSTIA estimations, there are no evident large-scale spatial pattern differences that indicate problems in representing the seasonal cycle. Instead, the differences, which are larger than 2 °C, seem to be more related to a misaligned occurrence of warm and cold eddies in comparison with the observations. Perhaps it would be easier to directly force RegCM5 with the GLORYS dataset, where mesoscale eddies should be in the proper location, because of the data assimilation.*

We thank the referee for pointing this out. We understand that the first version of the manuscript did not make it clear why we choose to use SYMPHONIE for forcing RegCM. There are two reasons for this choice: (1) We believe it is at least as reliable as GLORYS; (2) SYMPHONIE is the ocean model chosen for building an air–sea coupled regional climate model over SEA (also featuring RegCM), which is the topic of future studies. Naturally, reason (1) also motivates reason (2), and thus we focus here on developing reason (1).

First about the comparison with OSTIA. OSTIA is based on optimal interpolation, resulting – for our concern – in smoothing out most patterns smaller than roughly 100 km, all the more in

regions like SEA where cloudiness often limits the precision or visibility of observed patterns (Hihara et al., 2015; To Duy et al., 2022). The differences shown in Fig. A1 are thus more about SYMPHONIE resolving eddies where OSTIA simply shows a smoothed gradient, rather than a misalignment of equally contrasted structures.

Now, in comparison with GLORYS, let us develop on the confidence we place in the outputs from SYMPHONIE (in terms of biases, the comparison with GLORYS is comparable to that with OSTIA, not shown). In terms of setup, GLORYS is forced by ERA-Interim at the surface for our period, whereas SYMPHONIE uses the more updated analyses of the ECMWF. Also, SYMPHONIE uses a finer resolution, with 5 km against 8 km in the horizontal, and 60 levels against 50 in the vertical. Although it may not look like a great gap, Trinh et al. (2024) showed with SYMPHONIE that such a resolution change is very beneficial for the budget of heat, water and salt throughout the numerous narrow and shallow straits of the region. As for the assimilation conducted in GLORYS, SST is assimilated from Reynolds AVHRR-only SST and CORAv4.1 in-situ T-S profiles. The former is fairly less reliable than OSTIA (for the scales that interests us) in the sense that it also employs optimal interpolation but in addition relies on infrared data (which is a problem regarding the cloudiness of monsoonal climates), and the latter is a database of in-situ data which are too few in the SEA seas to concretely influence the submesoscale and mesoscale at the surface. As a result, there is in our opinion no reason to overtrust the submesoscale and mesoscale of GLORYS in SEA, and in particular, we think, as shown by Trinh et al. (2024) for the South China Sea, that downscaling GLORYS with SYMPHONIE over SEA using appropriate regional adaptations is a good strategy to get a refined daily description of SEA SST. Regional adaptations in SYMPHONIE, apart from the forcing data and resolution, include, e.g., the complex local bathymetry, land–sea mask and river forcing, and the consideration of internal tides, notably using an appropriate vertical advection scheme (internal tides greatly influence vertical mixing and surface characteristics in SEA, see Garinet et al. 2024).

Finally, in relation with the previous comment and our answer to it, choosing for GLORYS or SYMPHONIE should not affect how we compare the atmospheric configurations between each other, as long as the chosen forcing data is employed identically for all experiments. If we now consider that both GLORYS and SYMPHONIE make sense for forcing RegCM (as argued above), the choice may then depend on other considerations and objectives. Our choice for using SYMPHONIE is thus made in accordance with our broader objective of setting up an air–sea coupled regional climate model with RegCM and SYMPHONIE, without losing the generality of our conclusions regarding optimal RegCM configurations for the representation of air–sea fluxes in SEA.

The other half of the paragraph last quoted in the previous answer was modified to make it all clearer:

L163 "[...] the scale of 25 km (the horizontal resolution of our model). Optimal-interpolation-based SST datasets (e.g. the Operational Sea Surface Temperature and Ice Analysis, OSTIA, Good et al., 2020, or the Reynolds SST assimilated in the GLORYS12V1 global reanalysis, E. U. Copernicus Marine Service Information [CMEMS], 2020) generally capture the correct spatiotemporal variability on average, but they have been shown to overlook key mesoscale features in the region (To Duy et al., 2022). To address this limitation, our strategy is to force RegCM5 at the sea surface using output from SYMPHONIE (Marsaleix et al., 2008), a regional ocean model that we plan to couple with RegCM in future work as part of an air–sea coupled regional climate system – which underscores the particular relevance of choosing SYMPHONIE for the local long-term strategy. SYMPHONIE benefits from several years of expertise over the region, with a high resolution SEA configuration (Garinet et al., 2024) as well as different configurations throughout the northern SEA (Piton et al., 2021; Nguyen-Duy et al., 2021; To Duy et al., 2022; Herrmann et al., 2023, 2024; Trinh et al., 2024) showing good performances, in particular with regards to mesoscale processes and sea surface characteristics.. The SEA configuration covers the same domain as in Fig. 1, with a 5 km horizontal resolution and 60 vertical levels, resulting in an effective horizontal resolution of about 20 km, which ensures to resolve oceanic processes in the scale of RegCM's grid. SYMPHONIE runs from [...]."

**4. Need for clearer takeaways:** *Regarding the comparison of scores and rankings associated with the 36 experiments, it is difficult to identify which model aspect is more important or promotes more realistic results. Perhaps it would be better to build a figure that resumes Figure 3, which highlights the model aspects that mostly occur in the top 10 ranks. In this way, perhaps it would be easier to identify that experiment 12511 is the best performer.*

We understand that you would have liked us to highlight recurring schemes of the top 10 experiments. We did not think of this angle of analysis at first and thank you for your suggestion. The updated conclusion has one additional sentence answering your point:

L573 "The majority of the top 10 scheme combinations featured the Tiedtke cumulus convection, the UW-PBL, and the Xu–Randall cloud fraction schemes, while representing each option for radiative transfer and resolved-scale microphysics nearly equally. The Tiedtke cumulus convection scheme particularly outperformed others for modeling precipitation and [...]"

Moreover, we want to quote another updated paragraph located further in the conclusion which highlights somehow other schemes after comparing our results with another study focusing on land (after a comment from another anonymous referee), which also helps getting the big picture:

L632 "[...] in order to guide modelers seeking homogenous RegCM performance over the region, we can conduct as of now a brief comparison of our ocean-focused results with the land-only ones of Ngo-Duc et al. (2024). They notably identified four configurations with equivalent

aggregate scores, including three using Kain–Fritsch and one using Tiedtke. Our results indicate that Kain–Fritsch tends to overestimate oceanic monsoon signals in terms of precipitation, sea surface wind and latent heat flux, such that Kain–Fritsch configurations generally ranked in the bottom third of the ensemble. This supports favoring their top experiment that used Tiedtke instead. The Tiedke configuration highlighted in their study (i.e. 02510 using our notation) ranked third overall in ours while sharing the same PBL (UW-PBL), cumulus convection (Tiedke) and microphysics (SUBEX; i.e. *251*) as in the first and second ranks. This suggests that a balanced configuration may lie among these *251* combinations."

Lastly, we think that representing visually this recurring-scheme-in-top-10-configuration point fits the concept of graphical abstract. Therefore, we have worked on a graphical abstract presenting our topic, the region and study variables as well as the top 10 ranking with visual explanation on the schemes. This may still be changed before final publication but here is the current version:

[Figure]

**Figure 1.** Graphical abstract proposal.

**Specific comments**

*5. Line 182: Define all acronyms, like EBAF and IMERG.*

We agree with this comment and accordingly defined EBAF, IMERG, ERA5, ECMWF and OSTIA at their first mention. We chose not to define scheme acronyms, as commonly done within the RegCM community (see, e.g., Giorgi et al., 2023) where they are used more like proper nouns.

*6. In line 300, the authors mention that Tiedtke configurations "clearly" outperform the other configurations, but this is not easy to see.*

In the paragraph you refer to, the focus is made on sea surface wind speed (SSW). The sentence you point out reminds that Tiedtke configurations are written with a middle 5 ("∗∗5∗∗") and we can indeed see "clearly" that, except for 01511, the 11 ∗∗5∗∗ experiments perform all strictly better than others, with at least 1 score point gap with respect to the first non-Tiedke experiment for this variable, namely, 11410. We believe that confusion may come from losing the focus on SSW, i.e., on Fig. 3(d) (and not Fig. 3(e)) and thus we added a reference to this specific subfigure in the updated text. Moreover, the detail about the one score point gap (justifying the "clearly") in now included:

L307 "In particular, Tiedtke configurations (∗∗5∗∗) clearly outperform the others (Fig. 3(d)). The top 11 configurations for SSW indeed exclusively feature Tiedtke experiments, with at least a one score point difference relative to the first non-Tietdke experiment for this variable, namely 11410, ranking 12th (01511 is the only Tiedtke experiment which underperforms, standing at the second-to-last place)."

*7. In line 380, the authors highlight the weaker similarity along the Equator in Figure 4, but the figure lacks coordinates, which makes it difficult to follow the text.*

We agree with this comment and accordingly added the Equator to the maps of similarity in Fig. 4.

*8. The authors commonly refer to correlation coefficients, but there is no figure or table to see them, as in line 415.*

All the statistics employed to compute the scores are displayed in Appendix B, and these are also the ones we refer to in the text. To make it clearer in the updated version, we made explicit

mentions to the relevant appendix figures each time statistics were employed in the text (L426, L470, L472).

**References**

Garinet, A., Herrmann, M., Marsaleix, P., and Pénicaud, J.: Spurious numerical mixing under strong tidal forcing: a case study in the south-east Asian seas using the Symphonie model (v3.1.2), Geoscientific Model Development, 17, 6967–6986, https://doi.org/10.5194/gmd-17-6967-2024, 2024.

Giorgi, F., Coppola, E., Giuliani, G., Ciarlo, J. M., Pichelli, E., Nogherotto, R., Raffaele, F., Malguzzi, P., Davolio, S., Stocchi, P., and Drofa, O.: The Fifth Generation Regional Climate Modeling System, RegCM5: Description and Illustrative Examples at Parameterized Convection and Convection-Permitting Resolutions, Journal of Geophysical Research: Atmospheres, 128, e2022JD038199, https://doi.org/10.1029/2022JD038199, 2023.

Hihara, T., Kubota, M., and Okuro, A.: Evaluation of sea surface temperature and wind speed observed by GCOM-W1/AMSR2 using in situ data and global products, Remote Sensing of Environment, 164, 170–178, https://doi.org/10.1016/j.rse.2015.04.005, 2015.

To Duy, T., Herrmann, M., Estournel, C., Marsaleix, P., Duhaut, T., Bui Hong, L., and Trinh Bich, N.: The role of wind, mesoscale dynamics, and coastal circulation in the interannual variability of the South Vietnam Upwelling, South China Sea – answers from a high-resolution ocean model, Ocean Science, 18, 1131–1161, https://doi.org/10.5194/os-18-1131-2022, 2022.

Trinh, N. B., Herrmann, M., Ulses, C., Marsaleix, P., Duhaut, T., To Duy, T., Estournel, C., and Shearman, R. K.: New insights into the South China Sea throughflow and water budget seasonal cycle: evaluation and analysis of a high-resolution configuration of the ocean model SYMPHONIE version 2.4, Geoscientific Model Development, 17, 1831–1867, https://doi.org/10.5194/gmd-17-1831-2024, 2024.

---

## Author Comment (AC2)

**General overview**

*In this manuscript, the authors evaluate the performance of the RegCM5 regional climate model in simulating air–sea fluxes over the Southeast Asian seas. The model was run at a 25 km resolution for the year 2018 using 36 different combinations of physical parameterization schemes, selecting from multiple options for convection, microphysics, planetary boundary layer (PBL), radiation, and cloud fraction. Atmospheric forcing was provided by ERA5 reanalysis at 0.25° resolution, while sea surface temperatures were obtained from the high-resolution SYMPHONIE ocean model running at approximately 0.083° resolution. Model outputs, such as precipitation, surface radiation, latent and sensible heat fluxes, and sea surface wind speed, were evaluated against satellite and reanalysis data. A multi-criteria decision-making framework, incorporating 180 performance metrics across eight oceanic subregions, was used to rank the experiments. The results indicate that the top-performing configuration is a combination of the RRTM radiative transfer scheme, UW-PBL planetary boundary layer, Tiedtke cumulus convection, SUBEX resolved-scale microphysics, and Xu–Randall cloud fraction (identified as 12511, i.e. RRTM/UW-PBL/Tiedtke/SUBEX/Xu–Randall), with the Tiedtke cumulus convection scheme consistently outperforming others, particularly in simulating precipitation and wind. The findings highlight cumulus convection as the primary driver of model performance and suggest that the optimal physical parameterizations may vary depending on the variable of interest (e.g., precipitation vs. shortwave radiation). The manuscript is well written, logically structured, and easy to follow, making it a worthy candidate for publication in Geoscientific Model Development. However, there are some points that need to be clarified.*

Thank you for your careful reading, understanding and appreciation of our proposed manuscript. We carefully read your comments and answer them down below (line numbers refer to the updated manuscript).

**Comments**

*1. First, the authors' use of simulation results from __only one neutral year__ (2018) to evaluate the model's performance is not sufficiently convincing. A single-year simulation provides only one monthly and annual value per grid point for each variable, which introduces substantial uncertainty into the performance assessment due to the lack of statistical robustness. Furthermore, by excluding years influenced by major climate variability phenomena such as ENSO and IOD, the evaluation overlooks the model's capacity to simulate responses under extreme conditions, one of the key strengths of dynamical models. As a result, the findings may be overfitted to neutral conditions and may not adequately reflect the model's robustness or broader applicability across different climate regimes.*

Thank you for bringing this up. As a first answer, we want to highlight that this article represents for us one early brick of a broader initiative relating to evaluating and tuning RegCM over SEA, and that in this context, conducting experiments over several years including both neutral and non neutral conditions is definitely something we have in mind and will apply in the future. Yet, we think that this article's assessment based on a single year was necessary to reduce the ensemble of candidate configurations before going further with more experiments. Those future experiments should be conducted with a greater simulation period indeed, but also potentially address other aspects of performance (as also mentioned when addressing your second point below). In this study, we can only highlight the uncertainties bound to our protocol and the implications in terms of how to interpret and use our results. This is done in the conclusions of the new manuscript version:

L599 "Several limitations are yet to be mentioned. Indeed, our strategy targeted the seasonal cycle, assessed over a single year selected for its neutrality with respect to large-scale oscillations. Consequently, our findings are specific to that context, and no conclusions can be drawn about the performance of this study's top-performing configurations under non-neutral conditions, in terms of intraseasonal/inter-annual variability or more. While this work provides a basis for identifying a subset of promising configurations, additional experiments are needed to further refine the selection, notably involving longer simulations to conduct more comprehensive diagnostics."

*2. This study is highly valuable for advancing our understanding of air–sea coupling and for supporting the development of coupled models. However, in many practical applications, the accurate simulation of **precipitation and temperature over land is even more critical**. In fact, coupled models are still relatively uncommon, and most studies continue to rely on standalone RegCM without ocean coupling. Therefore, I suggest that the authors conduct a parallel analysis using the same model configurations over terrestrial subregions where high-quality observational data are available.*

We cannot disagree with you here, as precipitation and temperature over land have been the focus of the great majority of the regional climate community for the past two decades (e.g. Juneng et al., 2016; Ngo-Duc et al., 2017; Ngo-Duc et al., 2024). In this article, we aim to fill a gap in the literature about the performance of RegCM in SEA over the oceans, precisely considering the exclusive attention of the literature on land performance so far, hence largely disregarding ocean variables. Studying how configurations identified as optimal over the oceans actually perform over land is now legitimate, but this will be addressed in a standalone paper with more comprehensive performance evaluation of the model applied to a smaller subset of configurations (and with a longer simulation period). Nevertheless, at the moment, the parallel analysis you suggest can somehow be found with the last sensitivity experiments of

CORDEX-SEA by Ngo-Duc et al. (2024). Relating the results of the two studies (ours and that of Ngo-Duc et al., 2024) is done in the updated conclusion to give the big picture to interested readers:

L626 "Another approach within the same constraints would have been to fine-tune one single configuration we would have chosen based on previous research. However, previous studies featured significantly less physical options than those tested here, and focused almost exclusively on land performance. For example, Ngo-Duc et al. (2024) recently employed 0∗∗∗0 configurations to assess land temperature and precipitation (and to our knowledge, only 01∗10 experiments were tested in earlier works). As a result, our understanding of how RegCM performs across the full SEA domain was incomplete, and some recent options were never assessed despite yielding good results in the present study (e.g. RRTM and Xu–Randall). After the current paper, assessing RegCM's most updated schemes over land would be a valuable follow-up. Nonetheless, in order to guide modelers seeking homogenous RegCM performance over the region, we can conduct as of now a brief comparison of our ocean-focused results with the land-only ones of Ngo-Duc et al. (2024). They notably identified four configurations with equivalent aggregate scores, including three using Kain–Fritsch and one using Tiedtke. Our results indicate that Kain–Fritsch tends to overestimate oceanic monsoon signals in terms of precipitation, sea surface wind and latent heat flux, such that Kain–Fritsch configurations generally ranked in the bottom third of the ensemble. This supports favoring their top experiment that used Tiedtke instead. The Tiedke configuration highlighted in their study (i.e. 02510 using our notation) ranked third overall in ours while sharing the same PBL (UW-PBL), cumulus convection (Tiedke) and microphysics (SUBEX; i.e. ∗251∗) as in the first and second ranks. This suggests that a balanced configuration may lie among these ∗251∗ combinations. Our work thus serves as a prerequisite before embarking on any fine-tuning efforts from a relevant configuration. According to our findings, future fine-tuning efforts should first target the cumulus convection scheme, which was the primary driver of performance. Radiative transfer, PBL, and microphysics should follow as secondary priorities, while the cloud fraction algorithm warrants lower focus."

*3. __Using ERA5__ at the same resolution (0.25°) to force RegCM5 is valid and appropriate for a controlled physics sensitivity study, as done by the authors. However, in this setup, the added value of high-resolution spatial detail from the regional model cannot be fully realized.*

We do agree with you on this point. 25 km was chosen in agreement with the experiments made in the CORDEX-SEA community, who eventually conducts regional climate projections thereby limiting the possibilities of high resolution for computing resource concerns. A refinement of resolution should be opted for in the next phase of CORDEX-SEA (or in other future coordinated regional experiments), in particular alongside the upcoming CMIP7. As of now,

indeed, we cannot (and do not claim to at this stage) evaluate finer spatial climate patterns in the region, relative to ERA5 (although relative to most CMIP6 models, the resolution is still refined). This will have to wait for future studies following the regional community's plans.

A new paragraph opening up the perspectives of the study now include this consideration (together with ideas of the other anonymous referee):

L646 "The research proposed in this article also invites further exploration. For example, while we chose to force RegCM with a high-resolution SST field from SYMPHONIE in place of traditional, smoother SST datasets, we did not address the impact of this choice on the outputs of the model. How oceanic mesoscale eddies and meander impact the formation of clouds and precipitation in the area? We employed a 25 km horizontal resolution, so this may limit the impact of oceanic mesoscale in the atmosphere, but will this influence of SST become more critical with future resolution improvements? Indeed, with the upcoming seventh phase of the Coupled Model Intercomparison Project (CMIP; Dunne et al., 2024), resolution should increase in both global and regional climate models (including those we employ). More generally, how will our performance ranking evolve with those new resolutions?"

**References**

Juneng, L., Tangang, F., Chung, J., Ngai, S., Tay, T., Narisma, G., Cruz, F., Phan-Van, T., Ngo-Duc, T., Santisirisomboon, J., Singhruck, P., Gunawan, D., and Aldrian, E.: Sensitivity of Southeast Asia rainfall simulations to cumulus and air-sea flux parameterizations in RegCM4, Clim. Res., 69, 59–77, https://doi.org/10.3354/cr01386, 2016.

Ngo‑Duc, T., Tangang, F. T., Santisirisomboon, J., Cruz, F., Trinh‑Tuan, L., Nguyen‑Xuan, T., Phan‑Van, T., Juneng, L., Narisma, G., Singhruck, P., Gunawan, D., and Aldrian, E.: Performance evaluation of RegCM4 in simulating extreme rainfall and temperature indices over the CORDEX-Southeast Asia region, International Journal of Climatology, 37, 1634–1647, https://doi.org/10.1002/joc.4803, 2017.

Ngo-Duc, T., Nguyen-Duy, T., Desmet, Q., Trinh-Tuan, L., Ramu, L., Cruz, F., Dado, J. M., Chung, J. X., Phan-Van, T., Pham-Thanh, H., Truong-Ba, K., Tangang, F. T., Juneng, L., Santisirisomboon, J., Srisawadwong, R., Permana, D., Linarka, U. A., and Gunawan, D.: Performance ranking of multiple CORDEX-SEA sensitivity experiments: towards an optimum choice of physical schemes for RegCM over Southeast Asia, Clim Dyn, https://doi.org/10.1007/s00382-024-07353-5, 2024.

---

## Author Response (AR2)

**Dear Travis O'Brien,**

We sincerely thank you and our anonymous reviewers for accepting our paper for final publication. We have proceeded to the corrections you requested and provided all the files in the form. We hope that everything is complete and remain available for any more requests.

Best regards, Q. Desmet et al.